# Tree species identity and composition shape the epiphytic lichen community of structurally simple boreal forests over vast areas

**Julian Klein** [1,2]*, **Matthew Low** [1], **Göran Thor** [1], **Jörgen Sjögren** [3], **Eva Lindberg** [4], **Sönke Eggers** [1]

**1** Department of Ecology, Swedish University of Agricultural Sciences (SLU), Uppsala, Sweden, **2** Swedish Species Information Centre, SLU, Uppsala, Sweden, **3** Department of Wildlife, Fish and Environmental Studies, SLU, Umeå, Sweden, **4** Department of Forest Resource Management, SLU, Umeå, Sweden

* julian.klein@slu.se

**Data Availability Statement:** All data files are available from Zenodo (http://doi.org/10.5281/zenodo.3899847).

**Funding:** The Swedish Research Council Formas (Grant 942-215-57) to SE, JS, and EL. https://

## Abstract

Greatly simplified ecosystems are often neglected for biodiversity studies. However, these simplified systems dominate in many regions of the world, and a lack of understanding of what shapes species occurrence in these systems can have consequences for biodiversity and ecosystem services at a massive scale. In Fennoscandia, ~90% of the boreal forest (~21Mha) is structurally simplified with little knowledge of how forest structural elements shape the occurrence and diversity of for example epiphytic lichens in these managed forests. One form of structural simplification is the reduction of the number and frequency of different tree species. As many lichen species have host tree preferences, it is particularly likely that this simplification has a huge effect on the lichen community in managed forests. In a 40–70 years old boreal forest in Sweden, we therefore related the occurrence and richness of all observed epiphytic lichens to the host tree species and beta and gamma lichen diversity at the forest stand level to the stand's tree species composition and stem diameter. *Picea abies* hosted the highest lichen richness followed by *Pinus sylvestris*, *Quercus robur*, *Alnus glutinosa*, *Betula* spp., and *Populus tremula*. However, *P. tremula* hosted twice as many uncommon species as any of the other tree species. Stand level beta and gamma diversity was twice as high on stands with four compared to one tree species, and was highest when either coniferous or deciduous trees made up 40–50% of the trees. The stem diameter was positively related to lichen richness at the tree and stand level, but negatively to beta diversity. For biodiversity, these findings imply that leaving a few trees of a different species during forest thinning is unlikely as effective as combining life-boat trees for endangered species with an even tree species mixture.

## Introduction

Biodiversity studies are often performed in natural or near-natural ecosystems [1] with the aim to understand more about threatened species or ecosystems. Such systems are usually

formas.se/ The funders had no role in study design, data collection and analysis, decision to publish, or preparation of the manuscript.

**Competing interests:** The authors have declared that no competing interests exist.

complex in terms of the components and processes that shape biodiversity patterns [2–5]. However, extremely simplified systems resulting from anthropogenic impacts such as agriculture or forestry dominate the landscape in many regions of the planet [6]. These areas are often understudied with regard to the factors that shape the occurrence and diversity of the remaining species, which are commonly considered uninteresting [7]. Biodiversity patterns are still important in such systems for the provision of ecosystem services and ecosystem resilience, or for biodiversity conservation for its own sake [8, 9]. Considering the extent of these areas, changes in management that impact biodiversity could have profound changes in the number and diversity of living organisms across massive scales. In Fennoscandia, at least 90% of the boreal forest consists of tree plantations that are selectively grown and harvested at an ecologically young age for the boreal region (ca. 80–100 years [10]). This practice has replaced natural dynamics and has greatly reduced the amount and variety of forest structural elements [4], with profound negative consequences for biodiversity in Fennoscandia [11, 12]. One such structural element, the tree species composition [13], is simplified not only during planting, but also through forest thinning, when deciduous trees are removed to promote economically more beneficial coniferous species [14]. While the consequences of this simplification for species of conservation concern has been the focus of a number of studies [15–18], few have investigated how the occurrence and diversity of common species at different spatial scales relates to: (i) the presence of individual tree species, and (ii) the number and relative frequency of tree species at the forest stand level. While the importance for biodiversity of the latter can be evaluated using many different groups of organisms, the former is preferably evaluated with organisms that live directly on the tree [11].

Epiphytic lichens are one organism group where complete species surveys are rare [19–21]. In the context of this study they can be used as a biodiversity indicators [22, 23]. Forests that host a high lichen diversity are also expected to host a high diversity of other organism groups through a shared requirement for key structural elements (but see [24]). Although there is evidence that epiphytic lichen diversity in boreal forests is related to the tree species composition (e.g. number of tree species at the stand level [25] and density of hardwoods [26]), these studies either focus on natural forest systems or managed forests that include several forestry cycle stages in the study area. Complete tree level species inventories of epiphytic lichens in simplified systems such as managed boreal forests are extremely rare, and have until now not been performed in young managed forest stands (which in Fennoscandia constitute 60% of the managed forests and ~55% of the total forest area [27]). Research here has instead been limited to applied questions such as how introduced tree species [28], different dead wood types [29–31], or the type of stand replacement affect their diversity [32]. Thus, we still largely lack an understanding of the principle ecological mechanisms shaping epiphytic lichen diversity in these young managed forests. Specifically, we do not know the host tree preferences of many epiphytic lichen species, nor how forest structural elements such as the tree species composition affects epiphytic lichen diversity at the stand level. Increasing our knowledge of these ecological processes can help determine which forestry practices maintain highest epiphytic lichen diversity in structurally simple boreal forests.

To address this, we surveyed structurally simple, 40–70 year-old managed boreal forests in Sweden and used multi-species occupancy models to predict: (1) the probability that a given epiphytic lichen species will occur on a given tree species, and (2) the number of epiphytic lichen species hosted by a given tree species. We used a Bayesian hierarchical species accumulation model to determine the effect of (3) tree species richness and (4) tree species frequency, on beta and gamma lichen diversity. We also asked: (5) is there an optimal tree species composition that maximizes the beta and gamma diversity of epiphytic lichens, (6) how does average tree stem diameter relate to epiphytic lichen diversity, and (7) do these findings have

implications for biodiversity enhancing forestry methods such as life-boat tree retention [33] or mixed-wood forestry [34]?

## Material and methods

### Study site and data collection

The study site is located east of Uppsala in Sweden (59.84˚N, 17.96˚E; ~10 km radius), and part of a long-term experiment investigating the effects of forest thinning on biodiversity on land owned by the forestry company Holmen AB. The forest stands are typical for the region and primarily managed for wood biomass production with an age of 40–70 years. Being part of the boreal forest zone, the dominant tree species are Norway spruce *Picea abies* (L.) Karst. and Scot's pine *Pinus sylvestris* (L.). Birch *Betula* spp. is the dominant deciduous tree, followed by aspen *Populus tremula* (L.), alder *Alnus glutinosa* (L.) Gaertn. and oak *Quercus robur* (L.). On the established 59 plots (50 m radius), we surveyed a 10 m radius subplot for the lichen inventory, whose location was randomly chosen either at the centre or 25 m to the east or west of the centre of the plot (for details on plot placement, selection and forest metrics consult Klein et al., 2020). In May 2018, a lichenologist (GT) noted the presence of all lichen species within 2 m above ground on the trunk and the branches separately on every living tree (dead branches included) with a diameter at breast height (DBH) >10 cm. On each subplot, we used the time necessary to find, with a high likelihood, all species occurrences per tree. Because our survey method only examined the lowest 2 m of the tree stem and branches, it was important that this section of the tree was representative of the lichen community on that tree [35]. Previous studies from Swedish forests similar to ours indicate that the lowest 2 m on the tree accurately represents the lichen species pool on each tree [36, 37]. In 22 of the 59 plots commercial forest thinning occurred in the winter of 2017–2018; thus, only the remaining trees could be surveyed for lichens. However, since the inventory took place within four months of thinning, lichen occurrences would still be associated with the forest structure present before thinning [38]. Thus, the inventory provided us with species occurrence data for each tree on 59 subplots. For each tree we knew the tree species and DBH (used for the tree level model) and the subplot's tree species composition and average stem DBH (used for the stand level model). For details on lichen species determination and nomenclature consult Table A.2 in Klein *et al.* [24]. A reference collection of the majority of species observed in this study can be found at the UPS herbarium at Uppsala University, Sweden. The data set used for this study can be downloaded from https://doi.org/10.5281/zenodo.3899847. The data was originally created by Klein & Thor [39] and reused but not modified under the CC BY 4.0 license by us.

### Statistical analysis

Both the tree level and stand level models were implemented as Bayesian hierarchical models programmed in JAGS [40] using the package *rjags* [41] in R [42]. We scaled (Z-score normalisation) all continuous explanatory variables (DBH and % tree species) to increase model performance and facilitate interpretability of the results. We defined three different sets of initial values for all estimated parameters and latent values and ran the models until convergence (potential scale reduction factor < 1.1 [43]) and mixing (visual inspection) of the three Markov chain Monte Carlo chains (MCMC). We extracted parameter estimates at every $40^{th}$ step from a total of 50000 samples from each of the three MCMC chains. The analysis can be downloaded here: (10.5281/zenodo.5495894).

**Tree level model.** We used a multi-species occupancy model [MSOM; 44] adjusted for complete detection to model the occurrence of every lichen species on a tree (occurrence on

either the trunk or branches).

$$occurrence_{k,i,p} \sim \text{Bernoulli}(P_{k,i,p}) \tag{1}$$

The occurrence (presence = 1, absence = 0) of species $k$ on tree $i$ on subplot $p$ followed a Bernoulli distribution with the probability of occurrence $P$ as the distribution parameter (Eq 1). To determine the host tree species preferred by each lichen, we then explained the probability of occurrence of lichen species $k$ on tree $i$ on subplot $p$ with the host tree species identity, using a logit–link function and the id of subplot $p$ as the group effect. We included the host tree DBH as a covariate, because previous research has shown that the stem diameter is a crucial variable in explaining lichen richness [45]. In MSOMs, the unknown community means are used as priors in order to achieve estimates for rarely observed species (Bayesian shrinkage [44]). The unknown community means are in turn estimated using hyperpriors. For each epiphytic lichen species $k$, we used a Normal prior for the host tree species and DBH parameters, and a halfCauchy prior [46] for the standard deviation (SD) of the subplot group effect. We used vague Normal and halfCauchy hyperpriors for the mean and SD of the Normal priors and a Uniform hyperprior with limits 0 and 5 for the scale parameter of the halfCauchy priors. We then predicted probability of occurrence of all epiphytic lichen species for each tree species and an average tree using the posterior distributions of the model parameters for the host tree species identity. We calculated the sum of these occurrence probabilities across all lichen species to estimate tree level lichen richness. We then calculated the pairwise differences in lichen species richness among all host tree species (comparable to an ANOVA) and the probability that lichen richness on a given tree species is higher than that on another tree species. We visualized the results for species richness using *ggplot2* in R [47].

**Stand level model.**   The number of sampled trees per subplot varied from 6 to 46 (mean = 19) and because an increased number of sampling units leads to an increased number of species detections, species richness cannot be compared across subplots without correcting for this sampling bias using a species accumulation curve [48]. We therefore used the occurrence of all lichen species on either the branches or the trunk to produce species accumulated data and fit a species accumulation curve to every subplot. To do this, we virtually reordered the sampled trees and calculated the accumulated number of unique species $U_{i,p,s} = \sum_{i=1}^{i.max_p} u_{i,p,s}$

for every tree $i$ and subplot $p$, $s$ = 100 times. We then calculated the mean accumulated species richness $U_{i,p} = mean(U_{i,p,s})$ for every tree and subplot and produced a species accumulation curve for every subplot by fitting a modified Michaelis-Menten function (MMf) to this data. The asymptote of the original MMf depicts the expected number of lichen species per forest stand if all trees had been sampled (Gamma diversity), and the half saturation parameter represents the number of trees required to sample half of the expected stand gamma diversity. In the modified MMf, the half saturation parameter is replaced by true beta diversity [49] and the unit at which alpha diversity (number of lichens on an average tree) is evaluated (Details on the modified MMf are in the S1 File). In this study, we assume species accumulation would become asymptotic long before all trees per stand would be sampled because an average forestry stand in Sweden (4.5 ha [27]) of the type we study contains far less lichen species than trees (500–1000 trees per ha with an average stem diameter of ~20 cm [24]). In this study, the parameters of the modified MMf (Eq 2) are therefore a measure of expected between tree beta

diversity and stand level gamma diversity.

$$U_{i,p} \sim \text{Normal}(\mu_{i,p}, \sigma_p)$$

$$\mu_{i,p} = \frac{gamma.div_p * i}{beta.div_p - 1 + i} \tag{2}$$

In the application of the modified MMf, the accumulated number of lichens $U_{i,p}$ on tree $i$ on subplot $p$, followed a Normal distribution with the mean being a function of the modified MMf (Eq 2).

$$gamma.div_p \sim \text{Normal}(\mu.\gamma_p, \ \sigma.\gamma)$$

$$\mu.\gamma_p = \text{explanatory variables} \tag{3a}$$

$$beta.div_p \sim \text{Gamma}\left(\frac{\mu.\beta_p^2}{\sigma.\beta^2}, \ \frac{\mu.\beta_p^2}{\sigma.\beta}\right)$$

$$\log(\mu.\beta_p) = \text{explanatory variables} \tag{3b}$$

The estimated gamma and beta diversity on subplot $p$ are then modelled with a Normal and a Gamma distribution respectively (Eq 3). We chose the Normal distribution for gamma diversity because gamma diversity is here measure as the mean across 100 accumulation curves. We chose the Gamma distribution with a log-link for beta diversity to constrain beta diversity to $> 1$, since beta diversity cannot be lower than one (Eq 3b). We explained gamma and beta diversity simultaneously with the same set of explanatory variables to answer our study questions. To find out whether a forest stand's tree species richness was related to lichen diversity, we explained beta and gamma diversity in Eq 3 with the subplot's tree species richness (1, 2, 3 or 4 tree species), and calculated pairwise differences from the posterior distribution of the model parameters, in the same manner as described in the section above (*Tree level model*). To see if the proportion of the different tree species that make up the forest affect lichen diversity, we used the tree species composition on a subplot (only the trees that were surveyed for lichens) to explain beta and gamma diversity in Eq 3 in three different models. Here, we either used the frequency of Norway spruce, Scot's pine or deciduous trees as the explanatory variable. We chose a first order quadratic function for these three models because we assumed that 0% as well as 100% of a tree species in a forest stand could result in lower diversity than values in between. We then used these quadratic functions to identify an optimal tree species composition that maximises lichen diversity in a forest stand. We did this by: (1) calculating the percentage of the three tree species groups that was associated with the respective maxima in estimated beta and gamma diversity, and (2) by visualising the relationship between the actual tree species compositions found on the subplots and the associated diversity measures with ternary graphs. For the same reason as for the model for tree level lichen richness described above, we included the average DBH per subplot as a covariate in all models. We used the vague Gamma priors for the intercepts, vague Normal priors for the slopes and halfCauchy priors for the SD parameters. In order to verify that the model for beta and gamma diversity performed well, we predicted the accumulated species richness $U. pred_{i,p}$ on tree $i$ on subplot $p$ based on the estimated model parameters and checked if they follow the raw data $U_{i,p}$. We visualised the results using the functions *ggtern* [50] and *ggplot* in R. The number of tree species was not correlated with the tree species compositions.

## Results

We noted a total of 13,928 lichen occurrences of 117 lichen species on 1,101 trees in this study: Norway spruce = 5050 observations of 65 lichen species on 381 trees, Scott's pine = 7695 observations of 74 lichen species on 555 trees, Birch = 853 observations of 49 species on 116 trees, Aspen = 169 observations of 48 species on 27 trees, Alder = 125 observations of 25 species on 19 trees, and Oak = 36 observations of 25 lichen species on 3 trees. No red-listed lichens were found [12].

### Epiphytic lichens at the tree level

*Hypogymnia physodes*, *Ropalospora viridis*/*Fuscidea pusilla*, *Lepraria incana*, *Lecidea nylanderi*, *Parmeliopsis ambigua*, *Cladonia fimbriata*, and *C. coniocraea* were the most common species found, with the median probability of occurrence on an average tree ≥ 0.5 (Table 1). The *Cladonia* species were however less common on spruce and all of these species were less common on aspen. In addition to those species listed above, common lichen species (i.e. probability of occurrence ≥ 0.5) were: (i) for spruce: *Platismatia glauca*, *Hypogymnia tubulosa*, *Pseudevernia furfuracea*, *Parmelia sulcata*, *Violella fucata*, and *Lecanora pulicaris*, (ii) pine: *Vulpicida pinastri* and *V. fucata*, (iii) oak: *Phlyctis argena*, (iv) alder: *Lecanora pulicaris*, *V. fucata*, and *Micarea prasina*, (v) birch, none, and (vi) aspen: *Lecanora subrugosa* and *Lecidella elaeochroma* (Table 1). The number of lichen species that were uncommon-to-rare (probability of occurrence range: 0.2–0.01) were similar for all species except aspen (spruce = 24, pine and oak = 21, alder = 22, birch = 19, and aspen = 40; Table 1). Thirty-nine lichen species were very rare on all tree species (P(occurrence) < 0.01; S1 Table).

The host tree's identity had a strong effect on lichen richness per tree (Fig 1). The average number of lichen species was highest on spruce followed by pine, alder & oak, birch, and aspen in decreasing order (Table 2). The host tree DBH had a positive effect on the occurrence of an average lichen species and thereby on species richness (Median = 0.08, 95% CI = 0.03–0.14).

### Epiphytic lichens at the stand level

A higher number of tree species was clearly associated with a higher beta and gamma diversity, with stands consisting of four different tree species hosting nearly twice as many lichen species and double the beta diversity as monoculture stands (Fig 2). Tree species percentage had a clear quadratic effect on lichen diversity at the stand level (Fig 3a), with a peak diversity at around 40–50% for all three tree species groups (Norway spruce, Scot's pine and deciduous trees; Fig 3a). The peaks did not differ between the tree species groups in the case of gamma diversity, while peak beta diversity was at a higher percentage for deciduous than for spruce trees (Fig 3a). A ternary plot of tree species percentages revealed that an even mixture of the three groups resulted in the highest epiphytic lichen diversity (Fig 3b). The DBH had a positive effect on gamma diversity in the model with the number of tree species (Median = 2.15, 95% CI = 0.12–4.24), but only a slightly positive or no effect in the model with the tree species compositions (spruce: Median = 0.65, 95% CI = -1.21–2.55; pine: Median = 1.3, 95% CI = -0.8–3.25; deciduous: Median = 1.38, 95% CI = -0.83–3.58). The DBH had no effect on beta diversity at the stand level in the model with the number of tree species (Median = -0.01, 95% CI = -0.06–0.05), and a slightly negative effect in the model with the tree species compositions (spruce: Median = -0.09, 95% CI = -0.15 – -0.03; pine: Median = -0.05, 95% CI = -0.11–0.02; deciduous: Median = -0.05, 95% CI = -0.11–0).

**Table 1. Epiphytic lichen species observed in this study and their probability of occurrence on a certain host tree species as well as on a tree in average.**

| Species identity | Alder | Aspen | Birch spp. | Oak | Pine | Spruce | Average tree |
|---|---|---|---|---|---|---|---|
| *Acrocordia gemmata* | 0(0–0.01) | 0.01(0–0.11) | 0(0–0) | 0(0–0.04) | 0(0–0) | 0(0–0) | 0(0–0.02) |
| *Anaptychia ciliaris* | 0(0–0) | 0.02(0–0.23) | 0(0–0) | 0(0–0.02) | 0(0–0) | 0(0–0) | 0(0–0.04) |
| *Anisomeridium polypori* | 0(0–0.01) | 0(0–0.02) | 0(0–0) | 0.05(0–0.55) | 0(0–0) | 0(0–0) | 0.01(0–0.09) |
| *Bacidia arceutina* | 0(0–0.01) | 0.08(0–0.29) | 0(0–0) | 0(0–0.04) | 0(0–0) | 0(0–0) | 0.01(0–0.05) |
| *Biatora efflorescens* | 0.06(0.01–0.27) | 0.01(0–0.1) | 0.01(0–0.02) | 0(0–0.19) | 0(0–0) | 0.04(0.01–0.08) | 0.03(0.01–0.08) |
| *Bryoria capillaris* | 0(0–0.01) | 0(0–0.01) | 0(0–0) | 0(0–0.03) | 0(0–0) | 0.01(0–0.04) | 0(0–0.01) |
| *Bryoria fuscescens* | 0(0–0.06) | 0(0–0.02) | 0(0–0.02) | 0(0–0.19) | 0(0–0.01) | 0.24(0.14–0.36) | 0.05(0.03–0.08) |
| *Buellia griseovirens* | 0(0–0.02) | 0(0–0.03) | 0(0–0.01) | 0(0–0.11) | 0(0–0) | 0.01(0–0.02) | 0(0–0.02) |
| *Caloplaca cerina* | 0(0–0.01) | 0.01(0–0.1) | 0(0–0) | 0(0–0.04) | 0(0–0) | 0(0–0) | 0(0–0.02) |
| *Candelariella xanthostigma* | 0(0–0.01) | 0.01(0–0.1) | 0(0–0) | 0(0–0.04) | 0(0–0) | 0(0–0) | 0(0–0.02) |
| *Catinaria atropurpurea* | 0(0–0.04) | 0.01(0–0.11) | 0(0–0.02) | 0.1(0–0.71) | 0(0–0.01) | 0(0–0) | 0.02(0–0.13) |
| *Chaenotheca chrysocephala* | 0(0–0.02) | 0(0–0.02) | 0(0–0) | 0(0–0.11) | 0(0–0.01) | 0.01(0–0.03) | 0(0–0.03) |
| *Chaenotheca trichialis* | 0(0–0.02) | 0(0–0.02) | 0(0–0.01) | 0(0–0.09) | 0(0–0) | 0.01(0–0.02) | 0(0–0.02) |
| *Cladonia cenotea* | 0.07(0–0.32) | 0.02(0–0.09) | 0.2(0.12–0.3) | 0.04(0–0.5) | 0.37(0.3–0.45) | 0.01(0–0.02) | 0.13(0.09–0.21) |
| *Cladonia chlorophaea* | 0(0–0.01) | 0.01(0–0.11) | 0(0–0) | 0(0–0.04) | 0(0–0) | 0(0–0) | 0(0–0.02) |
| *Cladonia coniocraea* | 0.87(0.71–0.95) | 0.12(0.03–0.31) | 0.81(0.71–0.88) | 0.58(0.09–0.94) | 0.9(0.86–0.94) | 0.29(0.21–0.39) | 0.6(0.5–0.68) |
| *Cladonia cornuta* | 0.01(0–0.06) | 0(0–0.03) | 0.01(0–0.02) | 0(0–0.21) | 0.01(0–0.02) | 0(0–0) | 0.01(0–0.04) |
| *Cladonia digitata* | 0.14(0.02–0.44) | 0(0–0.04) | 0.18(0.1–0.29) | 0.35(0.03–0.87) | 0.48(0.39–0.58) | 0.02(0.01–0.04) | 0.2(0.12–0.3) |
| *Cladonia fimbriata* | 0.63(0.38–0.83) | 0.16(0.05–0.36) | 0.57(0.45–0.69) | 0.67(0.14–0.97) | 0.87(0.82–0.9) | 0.18(0.13–0.25) | 0.51(0.4–0.6) |
| *Cladonia pyxidata* | 0.01(0–0.09) | 0.02(0–0.13) | 0.01(0–0.03) | 0.01(0–0.35) | 0.03(0.01–0.06) | 0.01(0–0.02) | 0.02(0.01–0.08) |
| *Cladonia squamosa* | 0.01(0–0.1) | 0(0–0.02) | 0.01(0–0.05) | 0.01(0–0.34) | 0.02(0–0.04) | 0(0–0) | 0.01(0–0.07) |
| *Cladonia sp. phyllocladia PD red* | 0.14(0.05–0.31) | 0.18(0.05–0.41) | 0.08(0.04–0.14) | 0.04(0–0.49) | 0.06(0.04–0.09) | 0.24(0.17–0.32) | 0.13(0.08–0.22) |
| *Coenogonium pineti* | 0.14(0.04–0.38) | 0(0–0.02) | 0.1(0.04–0.18) | 0.23(0.01–0.8) | 0.03(0.02–0.06) | 0.17(0.09–0.27) | 0.12(0.05–0.23) |
| *Evernia prunastri* | 0(0–0.02) | 0(0–0.01) | 0(0–0.01) | 0(0–0.04) | 0(0–0.01) | 0.1(0.05–0.18) | 0.02(0.01–0.03) |
| *Fuscidea arboricola* | 0.01(0–0.08) | 0(0–0.03) | 0(0–0.01) | 0(0–0.13) | 0(0–0.01) | 0(0–0) | 0(0–0.03) |
| *Fuscidea pusilla/Ropalospora viridis* | 0.91(0.77–0.97) | 0.11(0.03–0.33) | 0.84(0.73–0.92) | 0.7(0.14–0.98) | 0.92(0.88–0.95) | 0.93(0.88–0.96) | 0.73(0.63–0.81) |
| *Graphis scripta* | 0(0–0.06) | 0.01(0–0.11) | 0(0–0.01) | 0(0–0.1) | 0(0–0) | 0(0–0) | 0.01(0–0.03) |
| *Gyalolechia flavorubescens* | 0(0–0.01) | 0.08(0.01–0.23) | 0(0–0) | 0(0–0.06) | 0(0–0) | 0(0–0) | 0.01(0–0.04) |
| *Hypocenomyce scalaris* | 0(0–0.03) | 0(0–0.04) | 0(0–0.01) | 0(0–0.16) | 0.02(0–0.04) | 0(0–0.01) | 0.01(0–0.04) |
| *Hypogymnia farinacea* | 0.01(0–0.06) | 0(0–0.04) | 0.01(0–0.03) | 0.01(0–0.19) | 0.01(0–0.02) | 0.01(0–0.03) | 0.01(0–0.04) |
| *Hypogymnia physodes* | 0.95(0.76–0.99) | 0.31(0.07–0.72) | 0.96(0.88–0.99) | 0.62(0.07–0.97) | 1(0.99–1) | 1(1–1) | 0.8(0.68–0.92) |
| *Hypogymnia tubulosa* | 0.01(0–0.07) | 0(0–0.02) | 0.01(0–0.02) | 0(0–0.12) | 0.03(0.02–0.05) | 0.74(0.64–0.82) | 0.13(0.11–0.16) |
| *Imshaugia aleurites* | 0(0–0.05) | 0(0–0.03) | 0(0–0.02) | 0(0–0.17) | 0.01(0–0.02) | 0.01(0–0.02) | 0.01(0–0.04) |
| *Lecanora albellula* | 0.01(0–0.09) | 0(0–0.04) | 0.01(0–0.03) | 0.01(0–0.23) | 0.48(0.39–0.56) | 0(0–0.01) | 0.09(0.07–0.13) |
| *Lecanora carpinea* | 0(0–0.02) | 0(0–0.02) | 0(0–0) | 0.04(0–0.55) | 0(0–0) | 0(0–0) | 0.01(0–0.09) |
| *Lecanora chlarotera* | 0(0–0.01) | 0.06(0–0.3) | 0(0–0) | 0.03(0–0.58) | 0(0–0) | 0(0–0) | 0.02(0–0.12) |
| *Lecanora hypoptella* | 0(0–0.03) | 0(0–0.03) | 0(0–0.01) | 0(0–0.16) | 0.03(0.01–0.06) | 0.01(0–0.02) | 0.01(0–0.04) |
| *Lecanora pulicaris* | 0.83(0.6–0.95) | 0.13(0.03–0.37) | 0.29(0.18–0.43) | 0.21(0.01–0.82) | 0.3(0.2–0.41) | 0.5(0.36–0.62) | 0.38(0.28–0.52) |
| *Lecanora sambuci* | 0(0–0.01) | 0.01(0–0.1) | 0(0–0) | 0(0–0.04) | 0(0–0) | 0(0–0) | 0(0–0.02) |
| *Lecanora subintricata* | 0(0–0.03) | 0(0–0.03) | 0(0–0.01) | 0(0–0.11) | 0.01(0–0.02) | 0(0–0.01) | 0(0–0.03) |
| *Lecanora subrugosa* | 0(0–0.03) | 0.9(0.55–0.99) | 0(0–0) | 0.18(0–0.81) | 0(0–0) | 0(0–0) | 0.18(0.1–0.29) |
| *Lecanora symmicta* | 0(0–0.04) | 0(0–0.03) | 0(0–0.01) | 0(0–0.12) | 0.05(0.02–0.07) | 0(0–0.01) | 0.01(0–0.03) |
| *Lecidea nylanderi* | 0.64(0.35–0.85) | 0.08(0.02–0.25) | 0.67(0.53–0.8) | 0.38(0.04–0.87) | 0.98(0.96–0.99) | 0.87(0.79–0.92) | 0.6(0.5–0.72) |
| *Lecidea turgidula* | 0.01(0–0.08) | 0(0–0.03) | 0.01(0–0.03) | 0(0–0.24) | 0.42(0.31–0.53) | 0.03(0.02–0.06) | 0.08(0.06–0.13) |
| *Lecidella elaeochroma* | 0(0–0.03) | 0.51(0.08–0.86) | 0(0–0) | 0.02(0–0.39) | 0(0–0) | 0(0–0) | 0.1(0.02–0.18) |
| *Lepraria incana* | 0.51(0.19–0.8) | 0.03(0–0.12) | 0.72(0.57–0.84) | 0.84(0.31–0.99) | 0.62(0.49–0.74) | 0.96(0.93–0.98) | 0.61(0.49–0.7) |
| *Lepraria lobificans* | 0(0–0.02) | 0.01(0–0.09) | 0(0–0.01) | 0.04(0–0.53) | 0(0–0) | 0(0–0.01) | 0.01(0–0.1) |

*(Continued)*

**Table 1.** (Continued)

| Species identity | Alder | Aspen | Birch spp. | Oak | Pine | Spruce | Average tree |
|---|---|---|---|---|---|---|---|
| *Melanelixia glabratula* | 0(0–0.01) | 0(0–0) | 0(0–0) | 0(0–0.02) | 0(0–0) | 0.01(0–0.04) | 0(0–0.01) |
| *Melanelixia subaurifera* | 0(0–0.01) | 0(0–0.02) | 0(0–0.01) | 0.01(0–0.48) | 0(0–0) | 0(0–0.01) | 0(0–0.08) |
| *Micarea denigrate* | 0(0–0.06) | 0.02(0–0.11) | 0(0–0.02) | 0(0–0.24) | 0.23(0.17–0.3) | 0.01(0–0.02) | 0.05(0.03–0.09) |
| *Micarea misella* | 0(0–0.04) | 0(0–0.03) | 0(0–0.02) | 0(0–0.2) | 0.07(0.04–0.1) | 0(0–0) | 0.02(0.01–0.05) |
| *Micarea prasina agg.* | 0.67(0.39–0.87) | 0(0–0.03) | 0.15(0.08–0.28) | 0.38(0.03–0.89) | 0.13(0.08–0.2) | 0.39(0.27–0.52) | 0.29(0.19–0.41) |
| *Mycobilimbia epixanthoides* | 0(0–0.01) | 0.13(0.01–0.38) | 0(0–0) | 0(0–0.05) | 0(0–0) | 0(0–0) | 0.02(0–0.07) |
| *Mycoblastus sanguinarius* | 0.01(0–0.07) | 0(0–0.02) | 0(0–0.01) | 0(0–0.09) | 0(0–0.01) | 0(0–0) | 0(0–0.02) |
| *Ochrolechia microstictoides* | 0.07(0.01–0.29) | 0(0–0.05) | 0.04(0.01–0.09) | 0.02(0–0.35) | 0.06(0.03–0.09) | 0.06(0.03–0.09) | 0.05(0.02–0.11) |
| *Parmelia saxatilis* | 0(0–0.01) | 0.01(0–0.11) | 0(0–0) | 0(0–0.05) | 0(0–0) | 0(0–0.01) | 0(0–0.02) |
| *Parmelia sulcate* | 0.07(0.01–0.29) | 0.05(0.01–0.17) | 0.03(0.01–0.07) | 0.08(0–0.54) | 0.01(0.01–0.02) | 0.55(0.45–0.66) | 0.14(0.1–0.23) |
| *Parmeliopsis ambigua* | 0.53(0.16–0.83) | 0.12(0.02–0.38) | 0.65(0.49–0.79) | 0.17(0–0.78) | 0.95(0.92–0.98) | 0.49(0.36–0.63) | 0.5(0.39–0.63) |
| *Parmeliopsis hyperopta* | 0.01(0–0.08) | 0(0–0.04) | 0.01(0–0.03) | 0.01(0–0.29) | 0.13(0.09–0.18) | 0.01(0–0.03) | 0.03(0.02–0.08) |
| *Peltigera membranacea* | 0(0–0.01) | 0.05(0–0.27) | 0(0–0) | 0(0–0.05) | 0(0–0) | 0(0–0.01) | 0.01(0–0.05) |
| *Peltigera praetextata* | 0(0–0.01) | 0.05(0–0.22) | 0(0–0) | 0(0–0.05) | 0(0–0) | 0(0–0) | 0.01(0–0.04) |
| *Pertusaria borealis* | 0.03(0–0.18) | 0(0–0.01) | 0(0–0.02) | 0.02(0–0.31) | 0(0–0) | 0.01(0–0.04) | 0.01(0–0.07) |
| *Phlyctis argena* | 0.07(0.02–0.21) | 0.14(0.04–0.35) | 0.01(0–0.02) | 0.57(0.07–0.98) | 0(0–0) | 0.24(0.13–0.36) | 0.18(0.07–0.28) |
| *Physcia adscendens/tenella* | 0(0–0) | 0.03(0–0.21) | 0(0–0) | 0(0–0.03) | 0(0–0) | 0(0–0.02) | 0.01(0–0.04) |
| *Physcia aipolia* | 0(0–0.01) | 0.02(0–0.16) | 0(0–0) | 0(0–0.04) | 0(0–0) | 0(0–0) | 0(0–0.03) |
| *Physconia distorta* | 0(0–0.01) | 0.06(0–0.3) | 0(0–0) | 0(0–0.03) | 0(0–0) | 0(0–0) | 0.01(0–0.05) |
| *Platismatia glauca* | 0.01(0–0.13) | 0(0–0.02) | 0.01(0–0.03) | 0.04(0–0.42) | 0.02(0.01–0.04) | 0.84(0.75–0.91) | 0.16(0.14–0.23) |
| *Pseudevernia furfuracea* | 0(0–0.06) | 0(0–0.03) | 0(0–0.01) | 0(0–0.1) | 0.02(0.01–0.04) | 0.71(0.61–0.81) | 0.13(0.11–0.15) |
| *Pseudoschismatomma rufescens* | 0(0–0.01) | 0.01(0–0.11) | 0(0–0) | 0(0–0.03) | 0(0–0) | 0(0–0) | 0(0–0.02) |
| *Scoliciosporum chlorococcum* | 0(0–0.05) | 0.02(0–0.12) | 0(0–0.02) | 0(0–0.19) | 0.01(0–0.02) | 0.01(0–0.02) | 0.01(0–0.05) |
| *Toniniopsis subincompta* | 0(0–0.01) | 0.03(0–0.16) | 0(0–0) | 0(0–0.04) | 0(0–0) | 0(0–0) | 0.01(0–0.03) |
| *Trapeliopsis flexuosa* | 0.03(0–0.19) | 0.03(0–0.12) | 0.05(0.02–0.11) | 0.03(0–0.48) | 0.36(0.28–0.45) | 0.01(0–0.02) | 0.09(0.07–0.17) |
| *Tuckermannopsis chlorophylla* | 0(0–0.03) | 0(0–0.01) | 0(0–0.01) | 0(0–0.08) | 0(0–0) | 0.16(0.08–0.25) | 0.03(0.01–0.05) |
| *Usnea dasypoga* | 0(0–0.01) | 0(0–0) | 0(0–0) | 0(0–0.03) | 0(0–0) | 0.01(0–0.03) | 0(0–0.01) |
| *Usnea hirta* | 0(0–0.05) | 0(0–0.01) | 0(0–0.01) | 0(0–0.04) | 0.01(0–0.02) | 0.21(0.13–0.32) | 0.04(0.02–0.06) |
| *Violella fucata* | 0.8(0.57–0.94) | 0.11(0.02–0.34) | 0.27(0.17–0.39) | 0.26(0.02–0.8) | 0.59(0.5–0.67) | 0.5(0.4–0.6) | 0.43(0.34–0.54) |
| *Vulpicida pinastri* | 0.17(0.03–0.51) | 0.02(0–0.1) | 0.12(0.06–0.22) | 0.24(0.02–0.78) | 0.65(0.55–0.73) | 0.49(0.38–0.61) | 0.29(0.21–0.41) |
| *Xanthoria parietina* | 0(0–0.01) | 0.12(0.01–0.46) | 0(0–0) | 0(0–0.03) | 0(0–0) | 0(0–0) | 0.02(0–0.08) |

The numbers show the median and the lower and upper limit of the 95% credible intervals. Here, only species with a median probability of occurrence ≥ 0.01 on at least one tree species are shown. The lichen species with a median probability of occurrence < 0.01 on all trees are in S1 Table. The nomenclature follows Nordin et al. [51].

## Discussion

We know that having more tree species in managed boreal forests increases carbon storage, wood biomass and human food production [8, 52], as well as the diversity of birds and ectomycorrhizal fungi [53]. Our results show that this relationship also applies to epiphytic lichen diversity. The diversity of epiphytic lichens in managed boreal forests was strongly dependent not only on the host tree species, but also on the mixture and frequency of tree species in the local forest stand. These results provide important knowledge on such relationships in structurally simplified young boreal forests, and have significant implications for forest management strategies where concurrent biodiversity preservation, such as life-boat tree retention or mixed-wood forestry, are practiced. While we have focused on lichens in this study, the results are possibly relevant for biodiversity management for a wider range of taxa. This is because

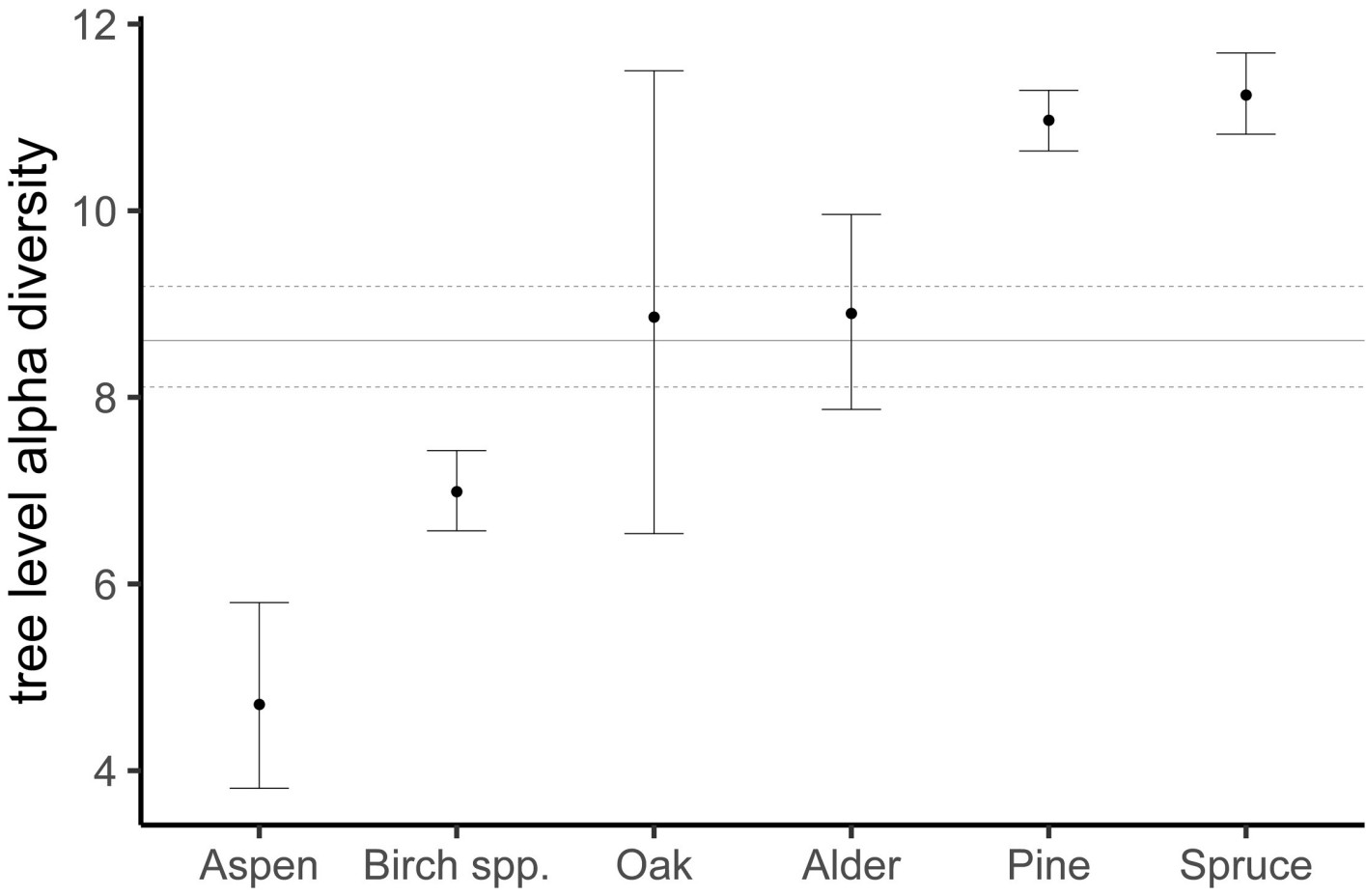

**Fig 1. Tree level epiphytic lichen richness (alpha diversity) per tree species with medians and 95% credible intervals.** The horizontal lines in the background shows averages across tree species. Complete pairwise comparisons are in Table 2. The data on which this figure is based can be downloaded from https://doi.org/10.5281/zenodo.3899847. The data was originally created by Klein & Thor [39] and reused but not modified under the CC BY 4.0 license by us.

**Table 2. The results of the pairwise comparison of the lichen richness on the different tree species (comparable to an ANOVA).**

|  | Aspen | Birch spp. | Oak | Alder | Pine | Spruce |
|---|---|---|---|---|---|---|
| Aspen |  | ~100% | 99.7% | ~100% | ~100% | ~100% |
| Birch spp. | 1.17–3.25 |  | 93.6% | ~100% | ~100% | ~100% |
| Oak | 1.66–6.89 | -0.43–4.52 |  | 50.3% | 94.5% | 96.2% |
| Alder | 2.78–5.57 | 0.84–2.97 | -2.84–2.57 |  | ~100% | ~100% |
| Pine | 5.16–7.2 | 3.51–4.42 | -0.54–4.46 | 1.02–3.08 |  | 87.3% |
| Spruce | 5.44–7.52 | 3.73–4.77 | -0.28–4.68 | 1.28–3.4 | -0.19–0.76 |  |

The lower left side shows the lower—and upper bound of the 95% credible intervals of the differences in richness on the tree species *row* vs. *column*, and the upper right side shows the probability that richness on tree species *column* is larger than that on tree species *row*. For example, spruce hosts between -0.19 and 0.76 more lichen species than pine with a probability of 95% and the probability that birch hosts more lichen species than aspen is 87.3%.

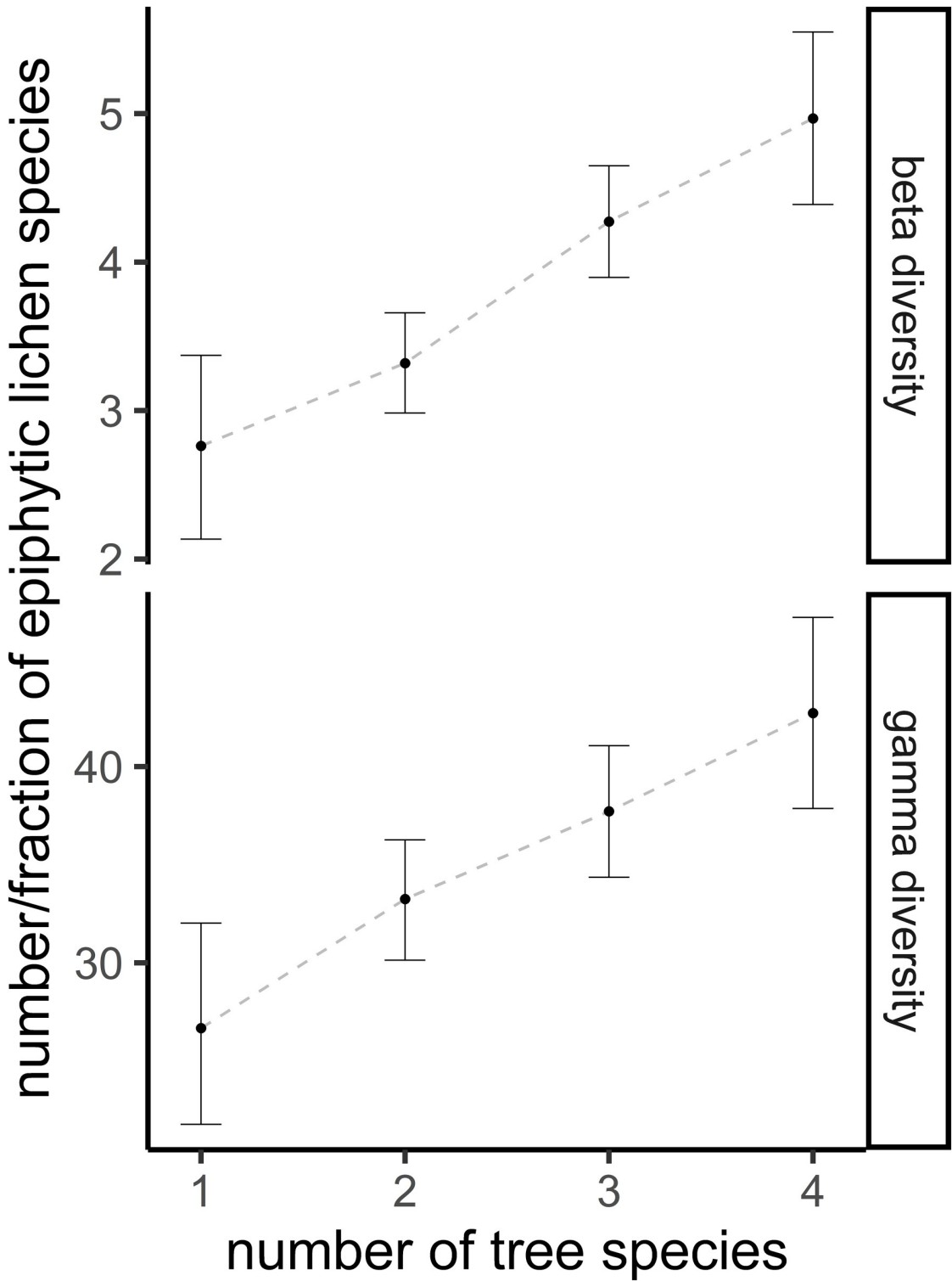

**Fig 2. Stand level beta and gamma diversity is modelled with the number of tree species in a subplot.** The predictions are the median together with the 95% credible intervals. Higher tree species richness was in all pairwise comparisons associated with higher lichen diversity with a posterior probability of 95%. The data on which this figure is based can be downloaded from https://doi.org/10.5281/zenodo.3899847. The data was originally created by Klein & Thor [39] and reused but not modified under the CC BY 4.0 license by us.

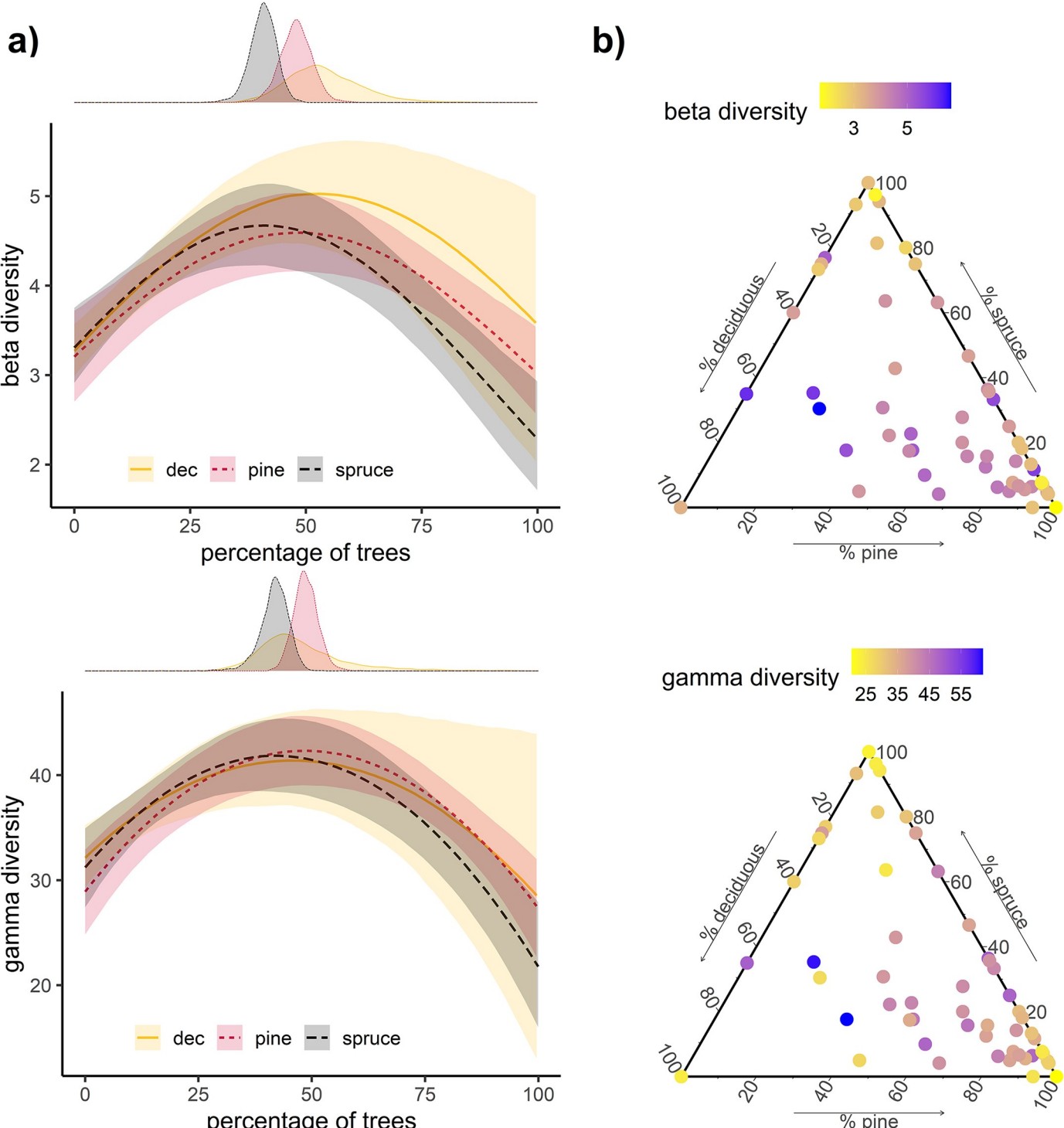

**Fig 3. a) Stand level beta and gamma diversity is predicted with the percentages of Norway spruce, Scot's pine or deciduous trees on a stand**. This was done with three separate models but they are here combined for visual comparison (e.g. at 40% spruce nothing can be said about po9the tree species composition of the remaining 60% of the trees). The predictions are medians with the 95% credible intervals. The probability densities of the percentage of a tree species which is expected to result in the highest number of lichen species is presented above for all tree species groups. **b) Ternary plot with the percentages of the tree species groups as the axis**. The colours show predicted beta and gamma diversity for the tree species percentages that occurred on the studied subplots. The data on which this figure is based can be downloaded from https://doi.org/10.5281/zenodo.3899847. The data was originally created by Klein & Thor [39] and reused but not modified under the CC BY 4.0 license by us.

lichens are regarded as an indicator taxon for habitat conditions suitable for other organism groups [22, 23].

## Epiphytic lichens at the tree level

No red-listed lichen species were observed in our surveys, suggesting that the structural elements which these threatened taxa require are largely absent from these structurally simplified young managed forests. What is most notable is the fact that almost no observed epiphytic lichen species sticks out in its distribution area or substrate preference. The lichen species which were common in this study usually turn out to be among the most frequent in lichen studies in boreal or nemoral forests [54]. The scarcity of calicioid species is consistent with earlier demonstrations that this taxa group is primarily associated with old-growth forests [e.g. 55]; many calicioid species are red-listed in Sweden [12]. While this has been observed in Estonia [56], the finding of *Arthonia vinosa* on Scot's pine is somewhat surprising for Sweden as this is an indicator of species rich broad-leaved deciduous forests [57]. The species identity of the host tree had a clear effect on the number of lichen species on that tree. Our results are supported by findings that Norway spruce is the most taxa-rich tree species in European boreal and boreo-nemoral forests (followed by the Scot's pine, the oak and other deciduous trees [58]). Even though lichen species on dead and live substrates were combined in their study, we can confirm these findings for lichens on live trees (the results for oak trees, however have to be interpreted with care as only three trees were inventoried). That spruce carried the highest number of lichen species was somewhat surprising given its low bark pH; as tree species with a higher bark pH (e.g. alder, oak and aspen) are usually those with the larger species pool of lichens [59]. One explanation for our results is that because we are looking at younger trees, spruce is a better host for lichens during this stage of forest development since the other tree species have not yet developed bark crevices and a chemistry that benefits many lichen species [60]. We could see some evidence of this in the strong positive effect of the tree diameter on lichen richness, with this relationship holding for all host species. However, stem diameter should correlate with the number of lichen species on a tree simply due to the species-area effect, or with greater age connected to a larger stem diameter, allowing a longer time for colonization [61]. Another possibility for the observed pattern is that Norway spruces have more branches reaching into the 0–2 m forest strata compared to other tree species, and increases the opportunity to survey branch-living lichen species [35]. But this potential biasing effect has not been found in previous studies of Swedish forests where the lowest 2 m on the tree accurately represented the species pool on each tree [36, 62].

## Epiphytic lichens at the stand level

Our study decisively demonstrates the importance of promoting forests with mixed tree species if the levels of epiphytic lichen diversity and associated ecosystem services and resilience are to be improved in young managed stands, which in Fennoscandia make up ~55% or ~10M ha of all forest land [27]. Forest stands with four different tree species are expected to host almost twice as many lichen species and twice the beta diversity as monocultures. That the tree species richness is strongly correlated with epiphytic lichen richness is a pattern known from tropical [63] and temperate forests [64]; however, these patterns have not been demonstrated in structurally simple managed boreal forests before. Our results are particularly relevant for boreal forests, where a reduction in tree species richness is a common result of management practices (such as thinning) that structurally simplify the forest to promote wood production. In boreal Europe, to include four or more different tree species in a stand would require including deciduous trees that are routinely removed during commercial thinning operations

[14]. Thus, the importance of deciduous tree retention for epiphytic lichen diversity, needs to be addressed in current forestry management in Fennoscandia. McMullin *et al.* [65] showed that the density of deciduous trees in general, but especially the occurrence of *Acer rubrum*, was positively related to high lichen richness in natural boreal forests of Nova Scotia, Canada. Unfortunately, we could not differentiate between different deciduous tree species as they did not occur in sufficient numbers to provide a clear picture of tree species-specific relationships to stand level lichen diversity. It is interesting to note, however, that even though aspen hosted the lowest number of lichen species per tree, aspen was a major contributor to between tree beta diversity, given that aspen trees hosted almost twice as many uncommon species as the other tree species in this study. Such relationships have also been suggested from natural boreo-nemoral forests in Estonia where aspen had the highest number of associated host specific lichen species and therefore the highest contribution to beta diversity among all tree species [66].

In this study, the Bayesian hierarchical modelling approach for species accumulation not only allowed us to investigate more complex relationships between predictors and the estimated gamma diversity, but also to extend the investigation of the influence of the predictors to beta diversity. In our study, this method enabled us to show that beta and gamma diversity was highest if a single tree species group constituted 40–50% of the trees in a stand and lowest at very low or very high percentages. This illustrates an important point, by showing that high tree species richness at the stand level does not automatically lead to higher epiphytic lichen diversity, but that the different tree species must also occur at relatively even frequencies within that forest stand. This has strong implications for the idea of biodiversity life-boat trees, where a few trees are retained in managed stands to allow taxa associated with those tree species to persist [33]. Based on our results, leaving a few trees of a different tree species during forest thinning that are expected to act as life-boats for biodiversity are unlikely to lead to the biodiversity levels that an even tree species mixture (i.e. as is found in mixed-wood forestry) would achieve. This, not saying that the retention of life-boat trees aiming specifically at preserving endangered lichen species [37] becomes obsolete. The maintenance of tree species richness to promote biodiversity in other taxa, thus needs to consider the possibility that the tree frequency in the forest stand is also critical. Such a concept, where tree species frequency is considered in addition to tree species richness, has been linked to increased ecosystem service levels and understory plants richness [52]. The generality of this patterns needs to be examined for other organism groups and ecosystem services and resilience to possibly motivate a change in forestry policy and practices to increase the frequency of especially deciduous trees (Current tree species composition: Norway spruce = 38%, Scot's pine = 43.1%, deciduous = 16.6% [67]). Deciduous trees in young Swedish managed forests are currently not only underrepresented in frequency but also in large stem diameter classes, which according to our results limits their contribution to stand epiphytic lichen richness. In fact, the stem diameter had a strong positive effect on gamma diversity, at least in the model with the tree species numbers. This well-studied pattern reappears in many tree environments where lichens are studied [37, 45]. The stem diameter was however only weakly positive in the models with the percentage of pine and deciduous trees, but had no additional effect in the model with the percentage spruce. Since the effect of DBH on richness is calculated for the case when the tree species' frequencies are at their mean and because these mean tree species frequencies were at (spruce) or near (pine and deciduous) the frequency that is associated with the maximal lichen richness, we can conclude from this result that a larger DBH does not increase lichen richness any further when tree species are at even frequencies. However the fact that the DBH was important in the model with tree species richness suggests that a larger DBH can lead to a higher lichen richness in forests with uneven tree species frequencies. It is also probable that

other factors associated with specific tree species, such as how they affect light penetration into the lower levels of the forest or how they affect each other's bark pH due to runoff connected to rain, also play a role in these relationships [24, 59]. That beta diversity was largely negatively affected by the average DBH, indicates that, as more and more different lichen species start to grow on trees with an increasing DBH, many trees also start to host the same species more frequently. This result is likely due to the fact that lichen colonisation on trees depends besides on the species' regional source pool and dispersal abilities also on coincidence [68]. To grow on a new tree, a lichen species either needs to be present in the immediate surroundings to spread vegetatively [69] or the different components of the lichen ecosystem need to meet simultaneously on the tree [70]. Which lichen species settles first on a young tree with a low DBH can therefore be coincidental, but with time and an increased bark area due to a higher DBH this colonisation becomes much less stochastic and more governed by e.g. dispersal distances along with stem size and age [61, 71, 72]. However, the fact that we noted a large number of epiphytic lichen species with a very low probability of occurrence, could indicate that the whole system we study still is in this pioneering phase. This is not surprising as the diversity of epiphytic lichens in boreal forests stabilizes only at a stand age of well above hundred years [68, 73].

## Supporting information

**S1 File. The hierarchical species accumulation model.**
(DOCX)

**S1 Table. Epiphytic lichen species observed in this study and their probability of occurrence on a certain host tree species as well as on a tree in average.** Here species with a median probability of occurrence < 0.01 on all trees are shown.
(DOCX)

## Acknowledgments

We are thankful to Aina Thor, Ola Hammarström and Simon Thor who assisted Göran Thor with great patience and attentiveness during the lichen survey. We want to thank the employees of the landowner Holmen AB who helped locating suitable forests for the lichen inventory. The data set used for this study can be downloaded from https://doi.org/10.5281/zenodo.3899847. The data was originally created by Klein & Thor [39] and reused but not modified under the CC BY 4.0 license by us. The code can be downloaded from 10.5281/zenodo.5495894.

## Author Contributions

**Conceptualization:** Julian Klein, Göran Thor, Jörgen Sjögren, Sönke Eggers.

**Data curation:** Julian Klein, Göran Thor.

**Formal analysis:** Julian Klein, Matthew Low.

**Funding acquisition:** Jörgen Sjögren, Eva Lindberg, Sönke Eggers.

**Investigation:** Julian Klein, Göran Thor, Sönke Eggers.

**Methodology:** Julian Klein, Göran Thor.

**Project administration:** Julian Klein, Sönke Eggers.

**Software:** Julian Klein.

**Supervision:** Sönke Eggers.

**Validation:** Matthew Low, Göran Thor.

**Visualization:** Julian Klein.

**Writing – original draft:** Julian Klein.

**Writing – review & editing:** Julian Klein, Matthew Low, Göran Thor, Jörgen Sjögren, Eva Lindberg, Sönke Eggers.

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
