## [Decision Letter · Decision Letter 0]

4 Jun 2021

PONE-D-21-14170

Tree species identity and composition shape the epiphytic lichen community of structurally simple boreal forests over vast areas

PLOS ONE

Dear Dr. Klein,

Thank you for submitting your manuscript to PLOS ONE. After careful consideration, we feel that it has merit but does not fully meet PLOS ONE’s publication criteria as it currently stands. Therefore, we invite you to submit a revised version of the manuscript that addresses the points raised during the review process.

We look forward to receiving your revised manuscript.

Kind regards,

Julian Aherne

Academic Editor

PLOS ONE

Additional Editor Comments:

I have received comments from four reviewers, all have noted the merit of your study and 3 of 4 have suggested minor revisions. I agree with this suggestion; nonetheless you are required to respond to the comments from all reviewers. Many of these comments refer to style, presentation, or clarification of written text; ultimately manuscript revisions in response to these comments will improve the impact of your manuscript.

Journal Requirements:

2. We noted in your submission details that a portion of your manuscript may have been presented or published elsewhere.

"The data used in this article has been used also for this article: https://www.sciencedirect.com/science/article/pii/S0378112720310963

The data was however used to answer a very unrelated question, e.g. we used entirely different explanatory variables in this article.

We therefore do not think this constitutes dual publication."

Reviewers' comments:

Reviewer's Responses to Questions

**Comments to the Author**

1. Is the manuscript technically sound, and do the data support the conclusions?

Reviewer #1: Yes

Reviewer #2: Partly

Reviewer #3: Yes

Reviewer #4: Yes

2. Has the statistical analysis been performed appropriately and rigorously? 

Reviewer #1: Yes

Reviewer #2: Yes

Reviewer #3: Yes

Reviewer #4: I Don't Know

3. Have the authors made all data underlying the findings in their manuscript fully available?

Reviewer #1: Yes

Reviewer #2: Yes

Reviewer #3: Yes

Reviewer #4: Yes

4. Is the manuscript presented in an intelligible fashion and written in standard English?

Reviewer #1: Yes

Reviewer #2: Yes

Reviewer #3: Yes

Reviewer #4: No

5. Review Comments to the Author

Reviewer #1: overall this paper is well written and researched; I have just a few concerns, listed below

some short detail justifying/explaining the sampling design would be nice; why E and W shifts only (e.g not also N and S?)

I am a bit puzzled by the choice of sampling at the very base of trees, since here the soil can exert a strong influence: I see e.g. Cladonia rangiferina, a terricolous lichen, in the list; please provide reasons for such a choice and add a discussion on this issue

why only the presence of the lichen species has been considered and abundance disregarded?

please give support to the use of Bernoulli, normal and gamma distributions to estimate occurrence, gamma and beta diversity, respectively

what is a lichen observation?

a short paragraph with itemized conclusions based on listed aims would be nice

please add a ref to species nomenclature followed in tab S1

Reviewer #2: This paper presents the results of a study on epiphytic lichen diversity in simplified boreal forest ecosystems. Among other aspects, the authors attempt to verify the relevance of tree species identity in determining the composition of epiphytic communities.

The idea of the work is interesting and the reference dataset very robust. the statistical approach is also more than adequate. However, I have several observations that relate in some ways to the background of the work, its generalisability and the presentation of the hypotheses to be tested. Below are some detailed observations:

In my opinion, the logical flow of the presentation of the work in the abstract should be constructed a little more clearly in order to make the reader understand the relevance of the subject matter. The authors begin by saying that simplified ecosystems are little taken into consideration, then they call into question an aspect that, in my opinion, seems to be linked to an increase in complexity in simplified forest systems (such as, for example, the presence/management of tree species that are more occasional than dominant). The context of the work is interesting but, at least in the abstract, a few more words should be spent to explain the presumed ecological processes that lead to the construction of the working hypothesis.

L35: I don't know if I really agree with the statement in the first line (which is also the basis for the preparation of the work framework) that biodiversity studies are mainly done in natural areas. This is also linked to another (forgivable) weakness of the work, namely the bibliographic support that is very much oriented towards boreal situations (here I agree: it is the ecosystem to which the authors refer) and which neglects an important body of literature on lichenology, but not only, that deals precisely with the study of diversity in areas that are also deeply altered. I can think of dozens of works in this sense from the USA, boreal, temperate and Mediterranean Europe. Many of these use the same diversity descriptors as the article. I think that the authors, while obviously keeping the focus on their reference ecosystem, should make a greater effort to relate their results to what has been observed in other forest systems. This would considerably increase the interest of the work, in my opinion.

L38. Although I can clearly see what the authors are referring to, I think a few more words should be spent on describing and defining these simplified ecosystems

L46: I believe that for a large part of the world, 80-100 years is not such a young age (neither in terms of logging nor in ecological terms) for a forest system!

L58: I’m sorry, but I think that epiphytic lichens are a 'heavily understudied' group seems to me to be a very strong and unsupportable statement.

I follow the context, but in my opinion the hypothesis should be presented much more explicitly: first you refer to ecological processes (dispersal, competition, establishment of these lichens would come to mind then), afterwards you say you want to test on which tree species there are more of them. Of course, I'm all for that, but I don't see much analysis of the ecological process there.

Methods: There are some aspects which, I imagine due to the need to simplify the sample design, seem to me to be somewhat neglected. The question of the tree species itself is resolved without, for example, taking much account of the physical and chemical characteristics of the bark or the structure of the tree as a whole. I do not seem to have seen any considerations related to the distance between trees or more generally to spatial connectivity within and between plots.

L78-88. I understand that the various points are extremely connected, but 7 objectives for one job seems too many. Perhaps there is a risk of the reader losing the thread a little (I have lost it, sorry). Without necessarily abandoning them, I would make the hypothesis more synthetically clear and fluent and keep some of those points as methodological aspects or perhaps merge some of them.

L255-258: I might well agree, but this statement is really very speculative and does not follow from the results presented. In general I find some parts of the discussion not really deducible from the observed results.

L353: Perhaps, but not necessarily: the presence of a high number of species with low frequencies (thus 'rare' at least in the study area) is a rather well-known and widespread phenomenon and can be found in ecosystems at different stages of succession.

Reviewer #3: This manuscript (PONE-D-21-14170) presents the results of the original research. The article is well written, discussion, and conclusions are made based on experimental data. The methodology is clear and sufficient.

There are a few my specific comments that need to be addressed which will improve the research study:

1.Please should add the authors of all the Epiphytic lichen species in Table 1. . The scientific names of species should be italicized.

2.Line 268-270 in Discussion: “The finding of Arthonia vinosa on Scot’s pine is somewhat surprising as this is a signal species indicating species rich broad-leaved deciduous forests (Nitare,2019).”

Marmor et al. 2011 studied epiphytic lichen biota on Picea abies and Pinus sylvestris in Estonia. They stated that Arthonia vinosa on Picea abies and Pinus sylvestris. You should also consider this article in the discussion.

Marmor et al. 2011., Effects of forest continuity and tree age on epiphytic lichen biota in coniferous forests in Estonia. Ecological Indicators, 11, 5, 1270-1276.

Reviewer #4: The global land area in wildernesses, i.e., ecosystems dominated by natural processes, has dramatically declined over the past 75 years, coinciding with an unparalleled growth in human populations. That change has created the biodiversity crisis and concerns that a sixth major extinction event is imminent. Thus, there is growing recognition that biodiversity must be protected globally.

It is therefore relevant to ask how biodiversity can be optimized in managed boreal forests—a globally extensive biome. This manuscript offers some useful and new advice in this arena based on intensively measured data for epiphytic macrolichens from a forest stand representative of highly managed forests in Fenno-Scandinavia. Lichens are useful indicators of forest health and biodiversity--other works have demonstrated that management practices that promote lichen diversity are likely to promote diversity of at least some other taxa groups.

The key new information demonstrated by the authors is that tree species diversity and evenness of that diversity promotes alpha and gamma diversity of lichens in forest stands. While this may not be good news for the timber industry —where the most economically valuable species are normally planted in higher proportions than less economically valuable species—it is useful information and likely to be of interest to managers of public forests and reserves, policy makers, and environmentally conscious timber producers. As expected, with longer forest continuity, represented by increasing trunk diameter, beta diversity evens out over time as slow to disperse species spread out evenly over the stand.

Because of the value of this information to managers, and the relative uniqueness of the dataset, I believe the findings of the authors deserve publication. However, I truly hesitate in my recommendation as the prose is inordinately difficult to understand. It is not a matter of grammar or familiarity with the English language, that is all fine. Rather it is a tendency toward verbosity and a practice of packing each sentence with so many words that this reader consistently had trouble fathoming sentence meaning. A reader should not have to read sentences multiple times to understand them. Further, I cannot see forest managers having the patience to read it. It is a funny thing—I recall my outrage in the film ‘Mozart’ when Emperor Franz explains that Mozart has used too many notes and that the listener cannot hear that many—therefore the composer should simply take some out. In this case, my recommendation is literally the same—except I mean words instead of notes. With more concise prose, this manuscript will be understandable to a much broader audience. Please find some examples of improved prose in the edited ms. This is not a complex dataset or analysis-- there is no reason for the ms to be difficult to understand. And, I do hope the authors will not share Mozart's outrage at my advice. As I recall, Mozart gauged his emperor to be an imbecile.

A last request. I had some trouble following the myriad terms in the methods section associated with the Bayesian modeling and this is why I am not confident in my ability to fully assess the statistical analyses. Perhaps a sentence or two describing the meaning of the word 'prior' and the significance of the different kinds of priors would help those readers who are not Bayesian experts. Revision of the figure and table descriptions is also needed, again the prose is hard to understand. Once I figured out the meaning of the text, I found the tables and figures relevant and useful.

6. PLOS authors have the option to publish the peer review history of their article (what does this mean?). If published, this will include your full peer review and any attached files.

Reviewer #1: No

Reviewer #2: No

Reviewer #3: No

Reviewer #4: No

---

## [Author Response · Author response to Decision Letter 0]

2 Sep 2021

Dear Julian Aherne,

We appreciate the opportunity to send a revised version of our manuscript “Tree species identity and composition shape the epiphytic lichen community of structurally simple boreal forests over vast areas” to PlosONE. The reviewer’s comments helped us to rewrite parts of the manuscript where the text or information was not sufficient for the reader to follow our work. We are grateful for their input and hope that the changes we have made and the answers we provide below are satisfying for publication in PlosOne.

1. The file manuscript is now fully formatted according to the PlosOne style guide. This includes also the naming of all files.

2. The same data for this article was used for a peer-reviewed and published article: https://www.sciencedirect.com/science/article/pii/S0378112720310963. In this article, we compared lichen richness and bird richness and how they are affected by the vegetation density. This published article does not answer any of the questions posed in the article submitted to PlosONE, e.g. we did not look at lichen data at the species level, nor beta diversity, nor did we analyse the effect of the tree species composition on lichen diversity. Therefore, all results of the article submitted to PlosONE are entirely new. 

3. We have adjusted the naming and in-text references of the supporting information according to PlosONE guidelines.

Concerning the reference list we have made the following changes:

We added the reference for the used nomenclature:

Nordin, A., Moberg, R., Tönsberg, T., Vittikainen, O., Dalsätt, Å., Myrdal, M., Snitt-Ting, D., Ekman, S., 2019. Santesson’s checklist of Fennoscandian lichen-forming and li-chenicolous fungi.

We added the reference suggested by reviewer 3 which states that A. vinosa has been observed on Scott's pine in Estonia: 

Marmor et al. 2011., Effects of forest continuity and tree age on epiphytic lichen biota in coniferous forests in Estonia. Ecological Indicators, 11, 5, 1270-1276.

See below our answers to the reviewers’ suggestions in black. Reviewer 4 commented directly in the manuscript. We therefore answer the concerns of reviewer 4 directly in this file (answers_to:reviewer_4.doc).

We thank you again for considering our manuscript for PlosONE.

On behalf of all authors,

Julian Klein 

Reviewer #1: overall this paper is well written and researched; I have just a few concerns, listed below

some short detail justifying/explaining the sampling design would be nice; why E and W shifts only (e.g not also N and S?)

-The randomly chosen subplots change between E centre and W because nest boxes had been installed to the east and to the west and since we measured the forest vegetation around these nest boxes it seemed natural to choose also these subplots plus the centre point for the lichen survey. We refer to a published article, which in detail describes this study design at this point in the manuscript. 

I am a bit puzzled by the choice of sampling at the very base of trees, since here the soil can exert a strong influence: I see e.g. Cladonia rangiferina, a terricolous lichen, in the list; please provide reasons for such a choice and add a discussion on this issue

-Bark and wood of all trees from the base and up to 2 m above the ground was inventoried. Other options like leaving the lowermost part would result in not including species mainly found near the ground (as Coenogonium pinetii and Anisomeridium polypori). Many lichen species are found on a specific substrate (e.g. calcareous rock, tree branches at seashores) but, randomly, species can occur on “wrong” substrate, e.g. the usually corticolous Vulpicida pinastri can be found on rocks. In this inventory, the usually terricolous species Cladonia rangiferina was found on bark.

why only the presence of the lichen species has been considered and abundance disregarded?

-We regard that the occurrence of lichens is a more conservative and reliable metric since abundance at the base of the tree, where the survey was performed, may not correlate with abundance on the whole tree. As we state in the manuscript, we can be confident that this detection issue is not a problem when using occurrence. Moreover, it would have been a much larger effort to collect abundance data. 

please give support to the use of Bernoulli, normal and gamma distributions to estimate occurrence, gamma and beta diversity, respectively

-We agree with reviewer 1 that more details are needed to understand the choice of the distribution. We added details that inform the reader more clearly, why a certain data distribution was chosen on L136 and L180 - L183.

what is a lichen observation?

-We agree that lichen observation is an unclear term. We therefore changed to: L207: “We noted a total of 13,928 lichen occurrences of 117 lichen…”

a short paragraph with itemized conclusions based on listed aims would be nice

-We see the point with such a list and did in fact use an itemised list to state our aims in the introduction. However we think the current way of presenting our conclusions is sufficient and regard it a matter of style and taste, whether such a list would be more suitable in the discussion than whole sentences as it is now. We here would like to let the editor decide on what is more suitable. 

please add a ref to species nomenclature followed in tab S1

-We have added the reference for the nomenclature in both lichen tables.

Reviewer #2: This paper presents the results of a study on epiphytic lichen diversity in simplified boreal forest ecosystems. Among other aspects, the authors attempt to verify the relevance of tree species identity in determining the composition of epiphytic communities.

The idea of the work is interesting and the reference dataset very robust. the statistical approach is also more than adequate. However, I have several observations that relate in some ways to the background of the work, its generalisability and the presentation of the hypotheses to be tested. Below are some detailed observations:

In my opinion, the logical flow of the presentation of the work in the abstract should be constructed a little more clearly in order to make the reader understand the relevance of the subject matter. The authors begin by saying that simplified ecosystems are little taken into consideration, then they call into question an aspect that, in my opinion, seems to be linked to an increase in complexity in simplified forest systems (such as, for example, the presence/management of tree species that are more occasional than dominant). The context of the work is interesting but, at least in the abstract, a few more words should be spent to explain the presumed ecological processes that lead to the construction of the working hypothesis.

-We have now added a sentence that introduces the scientific question asked in the article more properly.

L35: I don't know if I really agree with the statement in the first line (which is also the basis for the preparation of the work framework) that biodiversity studies are mainly done in natural areas. This is also linked to another (forgivable) weakness of the work, namely the bibliographic support that is very much oriented towards boreal situations (here I agree: it is the ecosystem to which the authors refer) and which neglects an important body of literature on lichenology, but not only, that deals precisely with the study of diversity in areas that are also deeply altered. I can think of dozens of works in this sense from the USA, boreal, temperate and Mediterranean Europe. Many of these use the same diversity descriptors as the article. I think that the authors, while obviously keeping the focus on their reference ecosystem, should make a greater effort to relate their results to what has been observed in other forest systems. This would considerably increase the interest of the work, in my opinion.

-Reviewer 2 has a point here with suggesting setting our work in a broader context, comparing our findings with those from other forest types globally. However, while we in fact do refer to comparable studies from the tropics, temperate forests L299, and the North-American boreal forest L306, we purpousely chose to focus on the boreal European forests as we wanted to give clear management suggestions for this ecosystem. In order to do this, we think it is important to also extensively discuss the particular ecosystem under management.

L38. Although I can clearly see what the authors are referring to, I think a few more words should be spent on describing and defining these simplified ecosystems

-We added “… such as agriculture or forestry…” to make it more clear what we refer to. Now: L38 “However, extremely simplified systems resulting from anthropogenic impacts such as agriculture or forestry dominate the landscape in many regions of the planet (Vitousek et al., 1997).”

L46: I believe that for a large part of the world, 80-100 years is not such a young age (neither in terms of logging nor in ecological terms) for a forest system!

-We agree with the reviewer that we were not specific enough: We added “…for the boreal region…” to make it more clear to the reader.

L58: I’m sorry, but I think that epiphytic lichens are a 'heavily understudied' group seems to me to be a very strong and unsupportable statement.

-We removed “heavily understudied” and now mention only that complete lichen surveys are extremely rare.

I follow the context, but in my opinion the hypothesis should be presented much more explicitly: first you refer to ecological processes (dispersal, competition, establishment of these lichens would come to mind then), afterwards you say you want to test on which tree species there are more of them. Of course, I'm all for that, but I don't see much analysis of the ecological process there.

-We understand the concerns of reviewer 2 about a lack of detailed ecological theory in the introduction. However, we specifically wanted to ask open questions instead of formulating a specific hypothesis in the introduction. This to address a broad readership, which likely includes practitioners interested in why our study is interesting for forest management. Instead, we chose to discuss the results with the help of ecological processes in the discussion. We do this extensively in the section “Epiphytic lichens at the tree level” and between L346-L360 where we discuss lichen ecology and biology to explain our findings to the reader.

Methods: There are some aspects which, I imagine due to the need to simplify the sample design, seem to me to be somewhat neglected. The question of the tree species itself is resolved without, for example, taking much account of the physical and chemical characteristics of the bark or the structure of the tree as a whole. I do not seem to have seen any considerations related to the distance between trees or more generally to spatial connectivity within and between plots.

-Reviewer 2 is correct that the spatial arrangement of trees within a subplot is a factor which could affect the occurrence of lichens on specific trees. We did not measure a variable which would allow us to take this into account in this study. However we discuss this now on L345-L347: “It is also probable that other factors associated with specific tree species, such as how they affect light penetration into the lower levels of the forest or how they affect each other’s bark pH due to runoff connected to rain, also play a role in these relationships (Hauck, 2011; Klein et al., 2020).” The spatial arrangement of the plots does in our opinion not affect the outcome since they are situated sufficiently far away from each other. We refer to a more detailed description of the study site on L102 where a map is provided as well as the distances between plots. Whether this information should also be available directly in this manuscript is according to us a matter of taste and up to the editor to decide. We are happy to provide more details if you the editor should decide this to be necessary.

L78-88. I understand that the various points are extremely connected, but 7 objectives for one job seems too many. Perhaps there is a risk of the reader losing the thread a little (I have lost it, sorry). Without necessarily abandoning them, I would make the hypothesis more synthetically clear and fluent and keep some of those points as methodological aspects or perhaps merge some of them.

-We understand that 7 aims might seem a lot. We have now rewritten this section and it is more readable now. However, the itemised numbering of the aims remains as we are convinced that it in fact makes the text more readable.

L255-258: I might well agree, but this statement is really very speculative and does not follow from the results presented. In general I find some parts of the discussion not really deducible from the observed results.

-In accordance with the reviewer’s concerns, we changed “likely” to “possibly” in order to weaken our statement. 

L353: Perhaps, but not necessarily: the presence of a high number of species with low frequencies (thus 'rare' at least in the study area) is a rather well-known and widespread phenomenon and can be found in ecosystems at different stages of succession.

-We agree with the reviewer, that this is a common phenomenon. However the extremely low frequency of some species is nevertheless in support this conclusion. But probably it is better if we draw this conclusion more weakly and therefore we changed to “could indicate” instead of “indicate”.

Reviewer #3: This manuscript (PONE-D-21-14170) presents the results of the original research. The article is well written, discussion, and conclusions are made based on experimental data. The methodology is clear and sufficient.

There are a few my specific comments that need to be addressed which will improve the research study:

1.Please should add the authors of all the Epiphytic lichen species in Table 1. . The scientific names of species should be italicized.

-We have changed this accordingly.

2.Line 268-270 in Discussion: “The finding of Arthonia vinosa on Scot’s pine is somewhat surprising as this is a signal species indicating species rich broad-leaved deciduous forests (Nitare,2019).”

Marmor et al. 2011 studied epiphytic lichen biota on Picea abies and Pinus sylvestris in Estonia. They stated that Arthonia vinosa on Picea abies and Pinus sylvestris. You should also consider this article in the discussion.

Marmor et al. 2011., Effects of forest continuity and tree age on epiphytic lichen biota in coniferous forests in Estonia. Ecological Indicators, 11, 5, 1270-1276.

-We were not aware of this fact and thank the reviewer for the contribution. We have added this to the discussion on L271.

Reviewer #4: The global land area in wildernesses, i.e., ecosystems dominated by natural processes, has dramatically declined over the past 75 years, coinciding with an unparalleled growth in human populations. That change has created the biodiversity crisis and concerns that a sixth major extinction event is imminent. Thus, there is growing recognition that biodiversity must be protected globally.

It is therefore relevant to ask how biodiversity can be optimized in managed boreal forests—a globally extensive biome. This manuscript offers some useful and new advice in this arena based on intensively measured data for epiphytic macrolichens from a forest stand representative of highly managed forests in Fenno-Scandinavia. Lichens are useful indicators of forest health and biodiversity--other works have demonstrated that management practices that promote lichen diversity are likely to promote diversity of at least some other taxa groups.

The key new information demonstrated by the authors is that tree species diversity and evenness of that diversity promotes alpha and gamma diversity of lichens in forest stands. While this may not be good news for the timber industry —where the most economically valuable species are normally planted in higher proportions than less economically valuable species—it is useful information and likely to be of interest to managers of public forests and reserves, policy makers, and environmentally conscious timber producers. As expected, with longer forest continuity, represented by increasing trunk diameter, beta diversity evens out over time as slow to disperse species spread out evenly over the stand.

Because of the value of this information to managers, and the relative uniqueness of the dataset, I believe the findings of the authors deserve publication. However, I truly hesitate in my recommendation as the prose is inordinately difficult to understand. It is not a matter of grammar or familiarity with the English language, that is all fine. Rather it is a tendency toward verbosity and a practice of packing each sentence with so many words that this reader consistently had trouble fathoming sentence meaning. A reader should not have to read sentences multiple times to understand them. Further, I cannot see forest managers having the patience to read it. It is a funny thing—I recall my outrage in the film ‘Mozart’ when Emperor Franz explains that Mozart has used too many notes and that the listener cannot hear that many—therefore the composer should simply take some out. In this case, my recommendation is literally the same—except I mean words instead of notes. With more concise prose, this manuscript will be understandable to a much broader audience. Please find some examples of improved prose in the edited ms. This is not a complex dataset or analysis-- there is no reason for the ms to be difficult to understand. And, I do hope the authors will not share Mozart's outrage at my advice. As I recall, Mozart gauged his emperor to be an imbecile.

A last request. I had some trouble following the myriad terms in the methods section associated with the Bayesian modeling and this is why I am not confident in my ability to fully assess the statistical analyses. Perhaps a sentence or two describing the meaning of the word 'prior' and the significance of the different kinds of priors would help those readers who are not Bayesian experts. Revision of the figure and table descriptions is also needed, again the prose is hard to understand. Once I figured out the meaning of the text, I found the tables and figures relevant and useful.

-We are very thankful for the reviewers extensive comments and suggestions. We have implemented most of them. As reviewer 4 gave suggestions on how to improve the manuscript directly in the text, we also decide to reply to these comments directly in the text. The file is termed: answers_to_reviewer_4.doc

---

## [Editor Report · Decision Letter 1]

6 Sep 2021

Tree species identity and composition shape the epiphytic lichen community of structurally simple boreal forests over vast areas

PONE-D-21-14170R1

Dear Dr. Klein,

We’re pleased to inform you that your manuscript has been judged scientifically suitable for publication and will be formally accepted for publication once it meets all outstanding technical requirements.

Kind regards,

Julian Aherne

Academic Editor

PLOS ONE

Additional Editor Comments (optional):

Well done. The revised manuscript fully addresses all comments from the four reviewers. I find it acceptable for publication. Thank you for your thorough submission.
---

## [Editor Report · Acceptance letter]

8 Sep 2021

PONE-D-21-14170R1 

Tree species identity and composition shape the epiphytic lichen community of structurally simple boreal forests over vast areas 

Dear Dr. Klein:

I'm pleased to inform you that your manuscript has been deemed suitable for publication in PLOS ONE. Congratulations! Your manuscript is now with our production department. 

Kind regards, 

on behalf of

Dr. Julian Aherne 

Academic Editor

PLOS ONE